# FROM IMAGES TO CONNECTIONS: CAN DQN WITH GNNS LEARN THE STRATEGIC GAME OF HEX?

## ABSTRACT

The gameplay of strategic board games such as chess, Go and Hex is often characterized by combinatorial, relational structures—capturing distinct interactions and non-local patterns—and not just images. Nonetheless, most common self-play reinforcement learning (RL) approaches simply approximate policy and value functions using convolutional neural networks (CNN). A key feature of CNNs is their relational inductive bias towards locality and translational invariance. In contrast, graph neural networks (GNN) can encode more complicated and distinct relational structures. Hence, we investigate the crucial question: Can GNNs, with their ability to encode complex connections, replace CNNs in self-play reinforcement learning? To this end, we do a comparison with Hex—an abstract yet strategically rich board game—serving as our experimental platform. Our findings reveal that GNNs excel at dealing with long range dependency situations in game states and are less prone to overfitting, but also showing a reduced proficiency in discerning local patterns. This suggests a potential paradigm shift, signaling the use of game-specific structures to reshape self-play reinforcement learning.

## 1 INTRODUCTION

In 2016, AlphaGo (Silver et al., 2016) became the first AI to beat professional Go players by combining Monte-Carlo tree search (MCTS) with neural network policy and value approximation and self-play training. Its approach has since been transferred to various other board games: AlphaZero (Silver et al., 2017) achieved superhuman strength in Chess and Shogi while Mohex-3HNN (Gao et al., 2018) has become one of the strongest AI for Hex. Behind these accomplishments lies a crucial observation: Despite the diversity of the board games, all these programs use convolutional neural networks (CNN) as the foundational architecture to approximate policy and value functions. CNNs excel at extracting spatial features from high-dimensional sensory inputs like images, enabling agents to effectively perceive and learn from their environment, a fundamental aspect in training RL agents. However they inherently exhibit relational inductive biases that favor spatial locality and translational equivariance. These biases align harmoniously for most positions in the game of

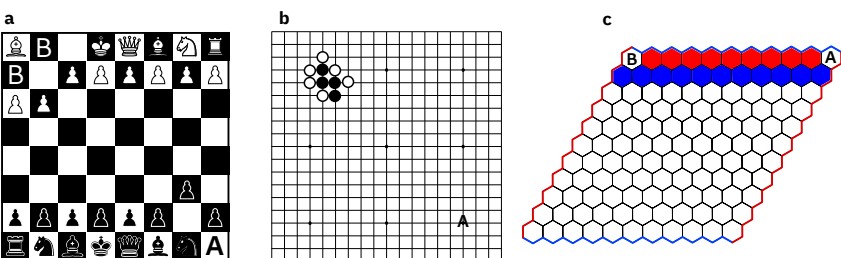

Figure 1: Non-local dependencies exist in many prominent board games. **a**, the bishop in the top-left corner is more related to the long distance square marked A, than the directly adjacent ones marked B. **b**, if there is a black stone at A, according to a typical Go pattern, white should not continue to play the ladder. **c**, if the tile at A is red, playing at B wins the game according to the Hex rules. If A is blue, playing at B is a wasted move.

Go where spatially local patterns dominate. As Figure 1 illustrates, non-local dependencies are very prevalent in other board games such as Chess or Hex. In these cases, the information-processing structure of CNNs does not align well with the structure of the game and learning becomes harder. This fact has already been observed by Silver et al. (2017) who noticed that AlphaGo Zero has a hard time understanding the "ladder" pattern which relates very distant areas of the Go board.

Good representation of the environment in Reinforcement Learning (RL) is crucial as it determines how effectively an agent can perceive and understand it, directly impacting the agents ability to make informed decisions and learn optimal policies. In contrast to CNNs, Graph Neural Networks (GNN) operate on graph representations instead of grid representations/images and can model complicated task-specific structure and relational inductive biases. In the case of Hex, cf. Figure 1c, a CNN using a grid-based representation primarily takes the information of the tiles close to (B) into account when predicting if (B) is a good move. A good graph representation on the other hand has edges closely connecting related parts of the board, such as (A) and (B) making it more likely that (A) is taken into account when predicting if the move at (B) is strong.

In this work, we use GNNs to overcome the mismatch between grid-based representation and the inherent structure of the task at hand. We propose Graph Deep Q-Networks (GraphDQN)[1] by integrating GNNs with RainbowDQN (Hessel et al., 2017) to capture non-local patterns and other task-specific relationships accurately while enabling self-play RL. To demonstrate the transformative potential of GNNs we also integrate GNNs with a methodology inspired by AlphaZero. We strategically choose the game Hex as a focal point study for this new approach. Hex serves as an ideal case study since it's a well known benchmark. Not only does Hex possess a classical grid-based representation but it also stands out as a "Shannon vertex switching game" (Gardner, 1961). This distinctive feature allows for an equivalent representation as a graph, adding a new level of complexity and interest to our investigation.

Our contributions can be summarized as follows:

- We introduce Graph Deep Q-Networks (GraphDQN), a combination of GNNs with DQN for use in self-play RL.

- To the best of our knowledge, we are the first to exploit the graphical structure of the game of Hex to train a distinctive Graph Neural network to play it.

- We empirically show that GNNs outperform CNNs trained similarly via DQN on game states with long range dependencies.

To this end, we proceed as follows: We start off by discussing related work. Then we introduce the game of Hex and show how to learn to play it using GNNs. Finally we conclude and provide avenues for future work.

## 2 RELATED WORK

The core contribution of this work is the combination of self-play reinforcement learning and graph neural networks and its application to the board game Hex. Graph Neural Networks (GNNs) are powerful tool for analyzing graph-structured data. They excel in tasks involving complex relationships, such as social network analysis, recommendation systems, and bioinformatics. GNNs utilize message-passing algorithms to capture intricate dependencies within graphs. Message-passing neural networks (MPNN), introduced by Gilmer et al. (2017), unify various previous graph neural network and graph convolutional network (Kipf & Welling, 2016) approaches into a message-passing framework on graphs. GraphSAGE (Hamilton et al., 2017) is an instantiation of a message-passing neural network that aims to generate high quality node embedding. We found GraphSAGE particularly suitable for function approximation in Hex due to it's straightforward message-passing structure and easy applicability. Many recent works leverage GNNs to improve deep reinforcement learning (DRL) methods (Almasan et al., 2022; Fathinezhad et al., 2023; Nie et al., 2023) and other domains such as visual scene understanding (Raposo et al., 2017), reasoning in knowledge graphs (Hamaguchi et al., 2017), quantum chemistry (Gilmer et al., 2017) or model-free reinforcement learning (Zambaldi et al., 2018). In self-play DRL, GNNs are still an under-explored topic.

---

[1]Our source code is available at `https://anonymous.4open.science/r/GNN_Hex-D90E`

Self-play reinforcement learning, describe a technique where an agent learns by playing against itself. It has garnered significant attention in recent years and has been widely adopted in the domain of artificial intelligence and reinforcement learning. One of the more prominent works in this area is AlphaGo (Silver et al., 2016) and its successor AlphaZero (Silver et al., 2017), which achieved groundbreaking successes in playing the board game Go and later Chess and Shogi. Building on this success, subsequent studies have explored the application of self-play in various domains, including other board games, video games, and robotics. A notable example is CrazyAra (Czech et al., 2020), a framework based on AlphaZero that extends to the chess variants Crazyhouse and Horde, as well as other games such as Hex and Darkhex (Blüml et al., 2023). Meanwhile, model-free approaches like DQN (Mnih et al., 2015) have been adapted to self-play RL, e.g. Neural-fictitious self-play (Heinrich & Silver, 2016). RainbowDQN (Hessel et al., 2017) combines several techniques, i.e., Prioritized Replay Buffers (Schaul et al., 2015), DoubleDQN (Van Hasselt et al., 2016) and DuelingDQN (Wang et al., 2016) to improve the performance and stability of DQN making the approach more applicable. Both CrazyAra and RainbowDQN have demonstrated remarkable performance in previous studies and will therefore be used in this work.

## 3  GAMES ON GRAPHS

Graph structures offer mathematical advantages with their visual clarity, providing a tangible representation for abstract concepts and aiding in theorem visualization. They find applications in graph theory, combinatorial analysis, number theory, and computer science. Their crucial role in modeling relationships between entities is vital for understanding various networks, from social interactions to data systems and ML approaches. Graphs serve as efficient data structures, simplifying complex relationships. In the context of games on graphs, they form a key component by combining combinatorics, game theory, and graph theory. A good overview on methods cab be found in the work by Fijalkow et al. (2023). In this work, we focus on applying Graph Neural Networks (GNNs) to self-play RL using such graph representation of games Waradpande et al. (2021).

**Shannon Vertex-Switching Games.** Shannon's Switching Game is a combinatorial game invented by Shannon with the goal of breaking down the connectivity within a network. Given a connected undirected graph $G = (V, E)$ with two distinct vertices $x$ and $y$, two players use *join* and *cut* actions in the graph in alternating turns. The *join* action removes a node from the graph while joining all neighbors of the removed node with a direct edge. The *cut* action removes a node without replacement. The *short* Player has the goal to connect the nodes $x$ and $y$ with a direct edge while the *cut* player wins by separating $x$ and $y$ into disconnected subgraphs making it impossible to connect them (Lehman, 1964). Shannon's Vertex-Switching Game is well known for being a generalization of Hex and Gale (Gardner, 1961) as such these games can be translated into Shannon Vertex-Switching Games unveiling logical equivalences between states, aiding strategic comprehension.

**The Game of Hex.** Hex is an abstract strategy board game invented by mathematician and philosopher Piet Hein and independently by mathematician John Nash. The game is played on a hexagonal grid of $n \times n$, typically with $n$ equals 11 or 13. Two players, each assigned a distinct color (often red and blue), take turns placing their pieces on the board with the goal of connecting their sides of the board with a continuous chain of pieces. The game offers a simple set of rules yet presents deep strategic complexity, often requiring players to think several moves ahead to anticipate and block their opponent's attempts to form a winning connection. Hex has been widely studied in the field of game theory and has served as an inspiration for various computer algorithms and artificial intelligence research due to its challenging gameplay and mathematical properties. Two examples of Hex boards are shown in Figure 1c and Figure 2a.

To represent a Hex board as a Shannon Vertex-Switching Game from the perspective of the red player, each uncolored tile and each of the red borders is represented with a distinct vertex, with edges connecting neighboring tiles. On alternating turns, red performs *join* actions, while blue performs *cut* actions on the resulting graph. Red wins if he can connect the border vertices $x$ and $y$ with a direct edge, while blue wins by cutting each connection between the two border vertices. In line with their respective goals, the red player is now called the short player while blue is called the cut player. Any Hex position can equivalently be represented from the perspective of the blue player, so that each Hex board has two equivalent Shannon-Vertex Switching Game representations. An example of a Hex graph with the corresponding board position is shown in Figure 2.

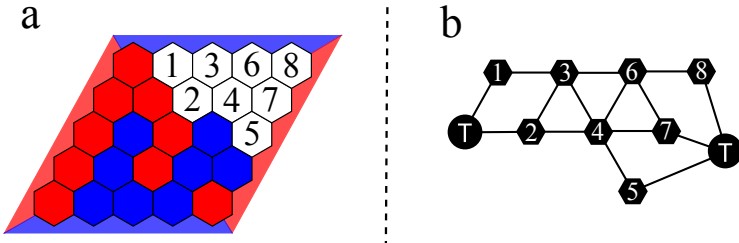

Figure 2: Example of a hex graph and it's corresponding board position. **a** A $5 \times 5$ Hex position. **b** The corresponding (red) Hex graph. The T-nodes represent the red sides while the other nodes are representing unclaimed tiles (The numbers indicate correspondence between tiles and nodes)

**Transferable CNN methods for Hex.** The most prominent existing Hex agents are the MoHex-based agents. MoHex was introduced by Arneson et al. (2010) and used Monte-Carlo Tree Search (MCTS, (Kocsis & Szepesvári, 2006)) to plan its moves. Several updates were published later on such as MoHex 2.0 (Huang et al., 2013), MoHex-CNN (Gao et al., 2017) and MoHex-3HNN (Gao et al., 2018). The last two are especially interesting because they integrate CNNs into MCTS similar to AlphaZero (Silver et al., 2017). CNNs that are conventionally used to approximate policy/value functions for board games are not fully convolutional. For instance, the AlphaGo Zero Silver et al. (2017) network includes a fully connected final layer to reach the desired output size and can thus only be designed for a specific input size. Gao et al. (2018) on the other hand only uses residual layers without pooling to preserve the board size. Similar to GNNs that can process graphs of any size and structure, Gao et al. (2018) architecture can deal with boards of any shape. Thus, it is especially suitable for comparison with GNNs in the context of Hex. Another approach to fully convolutional networks is the U-Net (Ronneberger et al., 2015a). It uses pooling layers together with up sampling to ensure the correct output size. We will use both Gao et al. (2018) and Ronneberger et al. (2015a) as baselines for comparison without GNN approach.

## 4 GNNs MEET SELF-PLAY RL

Graph Neural Networks (GNNs) offer significant promise in Reinforcement Learning (RL) due to their ability to model intricate relationships in non-Euclidean domains. In scenarios where agents interact in complex, graph-structured environments or those that can be effectively represented as such, GNNs exhibit prowess at capturing contextual information and propagating it across nodes. This empowers agents to make adaptive decisions based on diverse state representations, making GNNs a compelling approach for addressing the intricacies and dependencies inherent in RL problems. In this section, we propose two GNN based Self-play RL models: GraphDQN and GraphAra that judiciously integrates GNNs with DQN and AlphaGo respectively, to catalyze a paradigm shift in the domain of self-play RL.

**GraphDQN.** Deep Q-Networks (DQNs) (Mnih et al., 2015) was originally proposed as a single agent reinforcement learning algorithm and is still mostly used in single-agent tasks. However, a simple way to adapt DQN to self-play RL is by treating the opponent's moves as part of the environment. This means, that we compute the Q-target

$$Y^t = r_t - r_{t+1} + \gamma \max_a Q_{\theta'}(s_{t+2}, a) \tag{1}$$

from $s_{t+2}$, the state after the agent and his opponent have made a move. For choosing the opponent's move, it has been proposed to keep separate agents and average over the agents past behaviour (e.g. Neural Fictitious Self-Play (Heinrich & Silver, 2016)). However, for our purposes, comparing GNNs and CNNs on a game without imperfect information, we found that it was enough to just use the current version of the agent as the self-play opponent. To improve the stability and convergence speed of our self-play DQN training, we make use of various RainbowDQN (Hessel et al., 2017) techniques, mentioned in Section 2.

To test and evaluate the usability of GNNs in model-free self-play reinforcement learning, we decided to combine GraphSAGE (Hamilton et al., 2017), a GNN-based message passing architecture,

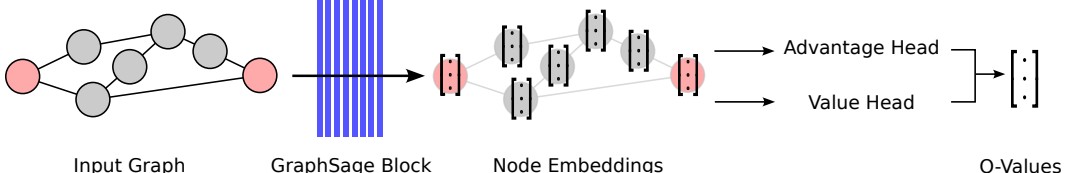

Figure 3: Architecture of the graph neural network used during the GraphDQN and GraphAra experiments. The network body uses a block of SAGEConv layers (Hamilton et al., 2017) to compute embeddings for each graph node. The advantage and value head implement DuelingDQN (Wang et al., 2016). Depending on if it is the short or the cut player's move, different parameters are used in the advantage head SAGEConv layers and the value head MLP. Advantages and values are added to compute the Q-values.

with RainbowDQN. The main goal here was to maintain the features of DQNs, such as training via self-play, but to take advantage of the graph representation.

For this, we replaced the convolutional layers with message-passing layers, within the DQN. The networks body is realized with a GraphSAGE message passing block that computes node embedding vectors for the input graph. These are passed to the two heads, the *advantage head* and *value head*.

In the advantage head, another GraphSAGE block is used with a final scaled tanh activation to compute advantages $A(s, \cdot)$ in the range $[-2, 2]$. To compute the position value $V(s)$ in the value head, the node embeddings are aggregated using symmetric aggregation operations. The mean, max, min, and sum aggregated node embeddings are concatenated into a single vector and a multilayer perceptron with final tanh activation is used to compute a position value in the range $[-1, 1]$. Finally, position value and advantages minus $\max_a A(s, a)$ are summed to calculate the output action-values. The overall architecture is shown in Figure 3, while an illustration of both heads can be found in Appendix C.

**GraphAra.** While RainbowDQN is still a very popular approach, AlphaGo (Silver et al., 2016) and later AlphaZero (Silver et al., 2017) have shown that model-based approaches are also well suited for self-play RL. To show that transformative capabilities of GNNs extend far beyond their synergy with RainbowDQN we combine GNNs with AlphaZero-inspired methodology, leveraging the adaptable framework of CrazyAra (Czech et al., 2020) which supports a variety of other games and chess variants, including the intricate terrains of Hex and its variants (Blüml et al., 2023). We use a similar GNN architecture to the one in GraphDQN Figure 3, with the only difference that the advantage head is replaced by a policy head and that the outputs of policy and value head are treated independently. Our approach thus differs from AlphaZero in that we have replaced the ResNet in AlphaZero with our GNN architecture. The algorithm is described in more detail in Appendix D.

## 5 EVALUATING GRAPHDQN AND GRAPHARA

**Experimental Setup.** A Graph Neural Network, a U-Net (Ronneberger et al., 2015a) and the transferable CNN by Gao et al. (2018) which will be further referred to as *Gao* were trained to play Hex using RainbowDQN. The GNN architecture is depicted in Figure 3 and constitutes 15 SageConv layers. The Gao network is a fully convolutional network consisting of ten residual blocks each compromised of two 3x3 convolutional layers. We modify the original Gao architecture (Gao et al., 2018) by removing policy and value head, as we only need the Q-head for RainbowDQN. Additionally, we do not use batch-normalization to ensure comparability with our GNN model that also does not use any normalization. The U-Net (Ronneberger et al., 2015a) has a total of three pooling layers thus processing the input of four different scales. The U-Net archtecture is illustrated in the Appendix B. All neural networks were implemented using pytorch (Paszke et al., 2019) with the GNNs making additional use of pytorch geometric (Fey & Lenssen, 2019). We ensure a fair comparison between the GNN and U-Net by choosing a similar amount of parameters (486974 vs 481329). The three networks were trained on 11x11 Hex boards for 110 hours on Nivida A100 40GB GPU. The GNNs process the Hex positions as graphs, receiving the set of edges and terminal node locations as inputs. In contrast, the U-Net processes Hex positions as a three layered grid, with two layers for red and blue stone locations and one layer indicating whose turn it is. The Gao

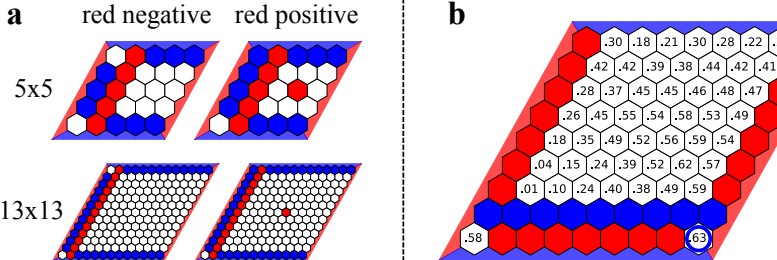

Figure 4: **a**, Long range dependency problems in Hex. Depending on the color of the tile at the top-left, red should (red positive) or should not (red negative) play at the bottom-left. Problems of the same shape are constructed for varied Hex sizes to evaluate the ability of CNNs and GNNs to solve long-range dependency problems. Results are shown in Table 1. **b**, Mistake made by the *Gao* CNN on the $9 \times 9$ *blue negative* long range dependency problem. With blue to move, the *Gao* CNN outputs the highest Q-value (.63) for the blue circled tile on the bottom right, which is a mistake.

Table 1: **GNNs perform better at long range dependency problems.** The GNN based GraphDQN makes significantly fewer mistakes on long range dependency problems than their CNN counterparts. Four long range dependency problems (*red positive*, *red negative*, *blue positive*, *blue negative*) similar to Figure 4 were created for each Hex size. The models, were evaluated on all of these problems by checking if they output the largest Q-value for the tile on the bottom left (or the corresponding graph node for the GNNs). Our approach is highlighted in **bold**.

| Agent | 8x8 | 9x9 | 10x10 | 11x11 | 12x12 | 13x13 | 14x14 - 25x25 | Sum of Errors |
|---|---|---|---|---|---|---|---|---|
| Gao | 1 | 1 | 1 | 2 | 2 | 2 | 22 | 31 |
| U-Net | 0 | 0 | 0 | 1 | 1 | 2 | 32 | 36 |
| **GraphDQN** | 0 | 0 | 0 | 0 | 0 | 0 | 1 | **1** |

network uses one additional layer for the location of empty tiles. Also, it indicates the color of each of the borders of the hex board by padding the input with red tiles on red borders and blue tiles on blue borders.

## 5.1 COMPARING GRAPH AND CONVOLUTIONAL NEURAL NETWORKS

**Long range dependencies.**   The primary reason to use GNNs in this work is to address the inherent challenges CNNs have when dealing with long range dependencies. To make this apparent, we make GNNs and CNNs solve specifically tailored Hex problems and evaluate their long range dependency potential. As shown in Figure 4, we create long range dependency problems on various Hex board sizes according to a predefined pattern. In *red positive* patterns, blue is threatening a connection on the left side via a single move on the bottom left. Thus, red solves this problem by playing bottom left himself. On the other hand, *red negative* patterns are similar, but the tile on the top-left is white instead of blue. Blue is thus not threatening a connection and playing on bottom-left is a mistake for red. This is a long-range dependency between the top-left and bottom-left tile. We create such patterns (and similar *blue positive* and *blue negative* patterns) for Hex board sizes from $8 \times 8$ to $25 \times 25$ and evaluate the GNN, as well as the two CNNs, Gao and U-Net, on these problems. Results are shown in Table 1. We find that CNNs exhibit significantly higher rate of errors. It's also worth noting that the errors made by U-Net can't solely be attributed to inability to transfer knowledge across board sizes, as U-Net also makes mistakes on Hex $11 \times 11$, the size it was trained on. In contrast, GNNs seem to have developed a general understanding of these long range dependency patterns, which remains robust even when dealing with exceptionally large board sizes.

**Board Size Transfer.**   We aimed to evaluate the ability of GNNs and CNNs to transfer knowledge to Hex board sizes unseen during training. Thus, we take our models trained via RainbowDQN on $11 \times 11$ Hex and set up matches between them on board sizes $8 \times 8$ to $15 \times 15$. The network agents

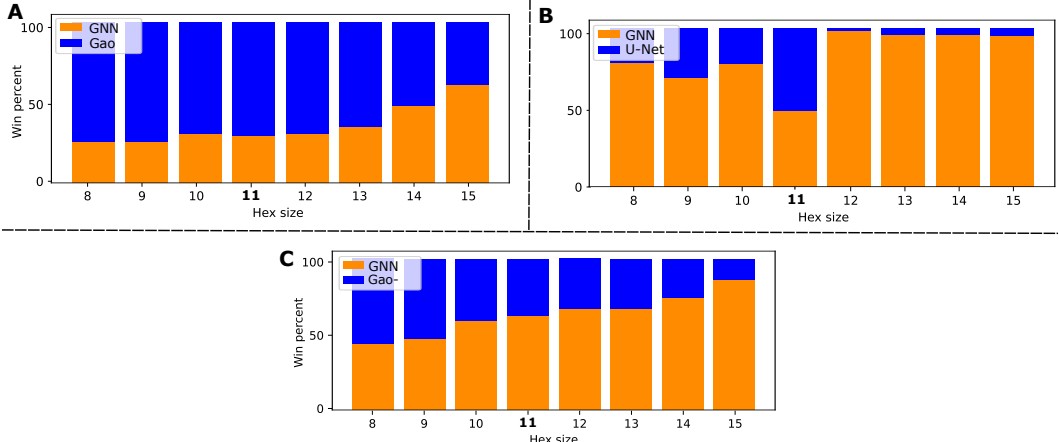

Figure 5: Performance of the Hex agents when evaluated on board sizes different from the training size. **A**, The Gao model is stronger than the GNN on smaller Hex sizes, but it becomes less effective when scaling up to a much larger board size. **B**, The U-Net performs much worse than the GNN when transferred to other board sizes. **C**, The Gao-network is trained and evaluated without padding the borders with red/blue tiles. This has a large negative impact on model performance, suggesting that border padding is key for training a strong CNN for Hex.

choose the moves with the highest predicted Q-values. To ensure diversity between the games we fix the starting move of each of the games played. On each board size, the agents play matches with each unique starting move and each player having the first move once. E.g. on $8 \times 8$ Hex there are 36 unique opening moves each of which is played by each of the agents in the first move for a total of 72 games played between the agents. Figure 5 shows the win rates of the agents on each board size. We find that the Gao network beats the GNN on most board sizes, but has trouble when transferring to much larger boards. The U-Net on the other hand only works well on the board size that it was trained on and performs much worse when transferring to other board sizes. Finally, we tested the impact of the border padding in the Gao model and thus trained a model Gao- that does not pad the borders of each hex board with red and blue tiles. We find that it performs significantly worse than model with border padding while showing the same trends for transfer between board sizes.

**Supervised learning.** In the next experiment, the GNN, the Gao model and the U-Net are trained to learn the policy of Mohex 2.0 and to predict the outcome of it's self-play games. The models were thus modified with a policy and value head. For the GNN this just means repurposing the advantage head as a policy head by replacing the final tanh activation with a softmax. For Gao, we use the policy and value head as described in Gao et al. (2018) and for the U-Net we add a feedforward value head and convolutional policy head similarly to the heads of the Gao model. All models are trained for 3000 training epochs using Adam (Kingma & Ba, 2014) optimization with a learning rate of $1 - e4$ and weight decay of $1e - 4$.

Figure 6 reveals that the U-Net overfits heavily on the 8000 Mohex 2.0 games in the training dataset, achieving high accuracy on the training set and low performance on the validation set. Similarly, the Gao model overfits heavily when predicting the games outcome (value sign accuracy), but still achieves the best validation accuracy on for predicting the policy. Only the GNN is able to generalize effectively to the validation dataset of 2000 Mohex 2.0 games achieving the highest validation value sign accuracy while not overfitting the training policy dataset.

## 5.2 Playing strength of GraphAra

To test the learning ability and to compare with conventional hex agents, we used our GNN approach in an AlphaGo-like framework as described in Section 4 and Appendix D. We first pre-trained an agent using supervised learning, based on a dataset of Mohex 2.0 games. The resulting agent *GraphAra-SL* was then further trained using self-play RL. The resulting model, *GraphAra-SP*, was compared in a round-robin tournament with Mohex 2.0 and GraphAra-SL. Each pairing played on

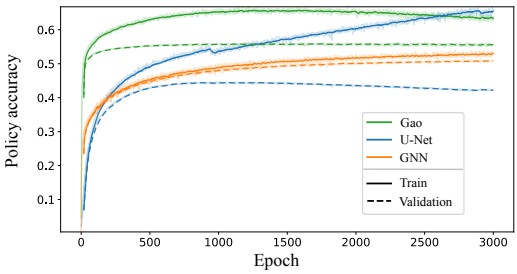 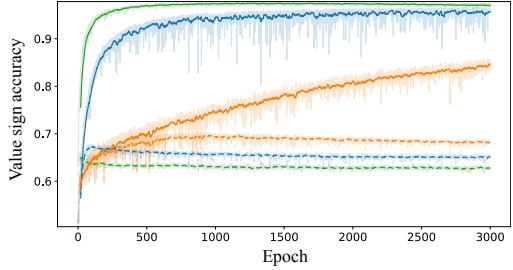

Figure 6: **GNNs are less prone to overfit in Hex.** When trained to imitate MoHex, the Gao architecture is best at predicting the MoHex policy (policy accuracy). In contrast, the U-Net architecture heavily overfits on the training dataset and achieves a low validation accuracy. The GNN converges significantly slower, but is the only model that does not overfit the MoHex policy. When predicting the outcome of each game (value sign accuracy), both CNN models heavily overfit on the training data which makes the GNN, which overfits less, the strongest model on the validation set.

Table 2: **GraphAra can play Hex competitively.** The table shows the results of a round-robin tournament played on all unique $11 \times 11$ Hex openings. The numbers represent the win percentages of the model from the left column vs the model from the top row and win ratios over 50% are highlighted in bold. The agents used are Mohes 2.0, a supervised trained model GraphAra-SL and GraphAra-SP, which was improved via self-play RL.

| Agent | GraphAra-SP | GraphAra-SL | Mohex-1k |
|---|---|---|---|
| GraphAra-SP | - | **79.55%** | **66.67%** |
| GraphAra-SL | 20.45% | - | **52.27%** |
| Mohex-1k | 33.33% | 47.73% | - |

all unique $11 \times 11$ openings with each agent starting once, resulting in 132 games played between each other. The results are shown in Table 2. Notably, GraphAra-SL, which is the GNN agent trained to imitate Mohex only using supervised learning, was able to beat its teacher Mohex-1k when played with 800 nodes per turn, while Mohex-1k using 1000 nodes. GraphAra-SP clearly benefited from the continued training, beating both GraphAra-SL and Mohex-1k by a significant margin.

## 6 DISCUSSION

The results in this paper suggest that given a suitable graph structure, there are several advantages of using GNNs over CNNs for function approximation in self-play reinforcement learning. In Hex, we identified long-range dependency problems that even sophisticated CNN based Hex agents struggle with (See Table 1). In contrast, the GNN was able to develop a general understanding of these long-range dependency problems through self-play reinforcement learning with zero prior knowledge. This understanding on a structural level is not broken even if the input at hand is very different from the training distribution (e.g. training on $11 \times 11$ Hex, testing on $24 \times 24$).

Given the impressive performance of GNNs on long-range dependency problems, we compared the playing strength of CNNs with GNNs. The direct comparison shows that the Gao architecture (Gao et al., 2018) trained with RainbowDQN beats the GNN trained via the same methods (See Figure 5). We attribute this shortcoming of the GNN to local patterns being more important in Hex play than previously thought and to the proficiency of CNNs in this area.

**Isomorph Hex Positions and Overfitting.** During the course of a Hex game, the initially uncolored Hex board becomes filled with red and blue tiles. If one were to treat the Hex board as an image, it's entropy would be $H = -\sum_{k=0}^{K-1} p_k \log(p_k)$ with $K = 3$ and $p_0, p_1, p_2$ being the proportion of uncolored, red and blue tiles respectively. With each move, the entropy of this image increases up to the point where there is an equal amount of red, blue and uncolored tiles. However, the Hex graph representation becomes one node smaller with each move played and at the point of

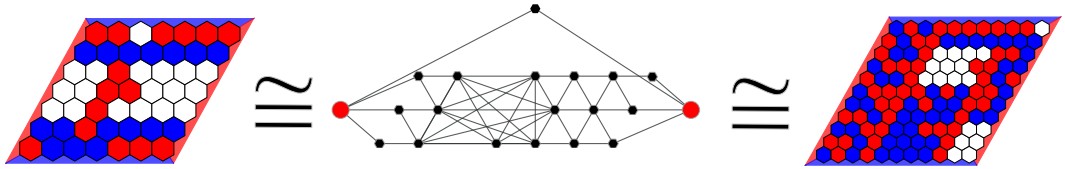

Figure 7: Very differently looking Hex boards even on different board sizes can have the same graph representation. On the left, there is a Hex position on the $7 \times 7$ board which is isomorph to the Shannon-Vertex switching game in the middle. However, the left one is also the graph representation of the very differently looking $11 \times 11$ board position on the right. Thus, the graph representation reveals that both Hex boards actually show an isomorph Hex endgame.

highest image entropy of the board representation it is only a third of it's original size. The reason for this discrepancy can be found in isomorphic endgame position. Larger boards contain a lot of information that is irrelevant for the evaluation of the position. The only thing that matters, in Hex, is which of the remaining uncolored tiles is connected to which border or which other uncolored tiles. An example of two positions with different board sizes, having the same graph representation, can be found in Figure 7. When we trained our neural networks to predict the evaluation of a Hex position based on a limited amount of data we found that the CNNs operating on the Hex board representation overfit a lot more than the GNN (Figure 6). Given that we did not use explicit regularization, the fact that the GNN overfits less has to be attributed to the different structure that the GNN is working on (Hex graph vs grid) and to the differences in how GNNs and CNNs process input data. The CNNs end up maximizing their training accuracy by learning the irrelevant layout of red and blue tiles in irrelevant parts of the board (such as in Figure 7 on the right). The GNN however operates on the graph structure which includes a lot less irrelevant information and can thus not easily overfit on the training dataset.

**Board Size Transfer.** GNNs have the inherent advantage that any GNN architecture can process graphs of any size, no matter the Hex board size they correspond to. In contrast, CNNs that include a fully connected layer are fixed to one board size. Ben-Assayag & El-Yaniv (2021) thus concluded that for the game Othello, GNNs are the best choice for board size transfer. Our experiments on Hex, comparing GNNs with fully convolutional neural networks reveal that the choice of CNN architecture plays a crucial part for it's ability to transfer knowledge between board sizes. Both the GNN as well as the Gao et al. (2018) CNN architecture turned out proficient at transferring knowledge between board sizes. However, the playing strength of the U-Net agent breaks down as soon as the domain changes to a Hex board size different from its training distribution.

**Limitations.** This work focused only on Hex and not without reason. Graph representations that are similarly efficient to Hex graphs are not known to exist for many other board games such as Go or Chess. Ben-Assayag & El-Yaniv (2021) have shown that one can even profit from the GNNs generalization abilities when the graph representation is not as efficient, such as in Othello. However, to apply the findings of this paper to another board game, one first needs to find good graph representations for this game, which is not always trivial, making this a limitation of our approach.

## 7  CONCLUSION

This work underscores a fundamental truth: Good structural representation is important and not every task is optimally represented by a stack of images. For some tasks and games, graphs can model the task inherent relationships more accurately. In Hex, we demonstrate that CNNs are prone to make mistakes related to non-local patterns due to the input representation. Similarly, unimportant areas of the Hex board input representation also led to strong signs of overfitting when CNNs were trained in a supervised manner. In contrast, GNN agents avoid mistakes on non-local patterns and overfit less in supervised training. These advantages can be attributed to graph networks being more accurate models of the relational structure of Hex, effectively capturing even non-local dependencies in image representation. With this work we have shown that GNNs can reasonably be used in self-play RL and how the resulting GraphDQN and GraphAra approach improve upon playing Hex. To conclude, the main message of our paper is to move beyond the one-size-fits-all approach and to use representations that fit the (game-)specific structures, problems and goals.

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

# A    PRUNING VERTICES IN HEX

It is commonly known that in Hex, there are moves on the board that can be pruned a priori, because they are irrelevant for the outcome of the game or strictly weaker than other moves (Henderson, 2010). Dead and captured cells can be colored immediately without changing the evaluation of a Hex position. A cell $c$ is dead if there exists no completion of the position (i.e. a filled Hex board that can be reached from the current position) where changing the color of $c$ changes the winner. A set of cells $C$ is captured by a player if he has a second player strategy on $C$, that renders all opponents moves in $C$ dead. Contemporary work such as Mohex 2.0 (Huang et al., 2013) precomputes local Hex board patterns in which cells are dead or captured and exploits them during search. For Shannon vertex-switching games, we can find general rules for finding dead or captured nodes based on properties of the neighborhood of each node. While playing out a game, this can be done efficiently, as only the neighbors of vertices removed in this turn have to be considered. Algorithm 1 finds and removes dead and captured nodes in a Shannon vertex-switching game. If the average amount of dead and captured nodes each turn is given by $d$, then Algorithm 1 will run in $O(d \cdot \mathbb{E}_{v \in G}[|N(v)|])$ time.

---

**Algorithm 1:** Removing dead and captured nodes in a Hex graph

---

**Input:** Nodes to consider $C$, undirected Graph $G$
**Function** deadAndCaptured$(C, G)$:
  $S \leftarrow \varnothing$
  **foreach** $v \in C$ **do**
    **if** $N(v)$ *fully connected* **then**
      $S \leftarrow S \cup N(v)$
      delete $v$ from $G$ ;                                  // $v$ is dead
      **continue**
    **foreach** $n \in \{N(v)_0\} \cup N(N(v)_0)$ **do**
      **if** $N(v) - \{n\} = N(n) - \{v\}$ **then**
        $S \leftarrow S \cup (N(v) - \{n\})$
        Add edges between neighbors of $v$
        delete $v$ and $n$ from $G$ ;        // short player captures $v$ and $n$
        **break**
    **foreach** $n \in N(v)$ **do**
      **if** $N(v) - \{n\}$ *fully connected and* $N(n) - \{v\}$ *fully connected* **then**
        $S \leftarrow S \cup (N(v) - \{n\}) \cup (N(n) - \{v\})$
        delete $v$ and $n$ from $G$ ;                // cut player captures $v$ and $n$
  **if** $S$ *not empty* **then**
    deadAndCaptured($S$,$G$)

---

## B  U-NET

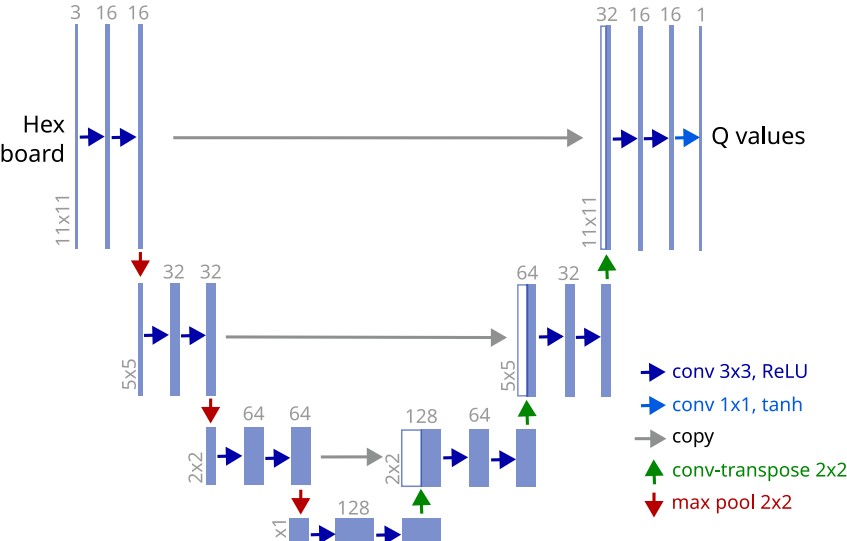

Figure 8: U-net architecture used to learn action-values for Hex positions. U-nets are fully convolutional neural networks that use upsampling / conv-transpose layers to ensure the correct output size, despite the use of pooling layers. Because they avoid fully connected layers, forward passes are possible with various input/output sizes using the same network. The illustration is inspired by Ronneberger et al. (2015b), Figure 1.

## C   ADVANTAGE AND VALUE HEAD

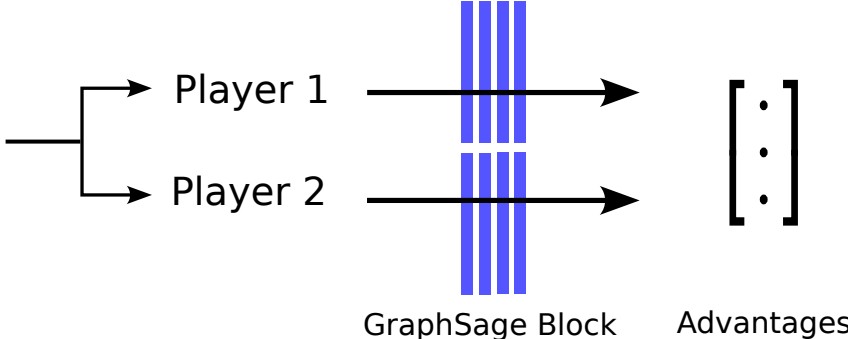

Figure 9: The Advantage Head in our pipeline (cf. Figure 3) is responsible for the estimation of an additional value, showing the difference between the value of the current state and the value of the next state. I measure the quality of improvement and quantifies the importance of each action relative to the current state. The output of the advantage head is a set of values, here displayed as a vector of values, one for each possible action in the current state. In our case, we use a number of SAGEConv layers within our GraphSage Block to calculate the values.

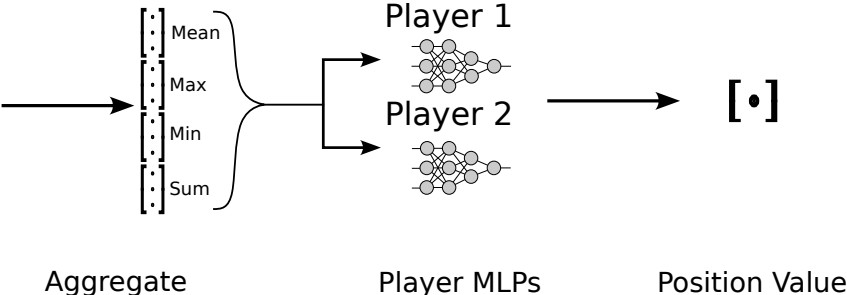

Figure 10: The Value Head in our pipeline (cf. Figure 3) gives us an estimation of the intrinsic value of being in a particular state. The output of the value head is a single value representing the quality of the current state. It captures the expected cumulative rewards from the state onwards. In our case we used aggregated features and specific MLPs for each player, given that each player has its own goal, to estimate the cumulative reward.

# D  ALPHAZERO MEETS GNN

The following algorithm shows the training process of GraphAra, based on AlphaZero. The loss function was taken from the original paper (Silver et al., 2016). In our experiments, a neural network was trained on game data from Mohex 2.0 (Huang et al., 2013) before being improves via self-play. Gere, we used the samples of 30,000 games per epoch ($T = 30,000$). If the new model won more then 50% of its games against the current best model, the updated model replaced the current best model in the learning process. For our training we used Three Nvidia A100 GPUs, one for GNN training and two for training data generation. We trained between 50 and 300 epochs, taking up 6 days of training. In the experimental section we used the model trained with the maximum number of epochs ($E = 300$). For the MCTS we used a computational budget of 800 simulations (or nodes). Our implementation is based on the work of Czech et al. (2020). Unlike classic AlphaZero, we use GNN instead of ResNet as the neural network $\theta$ and used a model trained on Mohex to begin with.

---

**Algorithm 2:** AlphaZero's Trainings Process (inspired by Blüml et al. (2023))

---

**Function** `AlphaZero`$(T, E, \ell, \theta)$:

> **Parameters:** number of games per epoch $T$, number of epochs $E$, loss function $\ell$, neural network $\theta$
>
> **repeat** $E$ **times**
>
> > $t \leftarrow 0$
> > $g \leftarrow 0$
> > **while** $g \leq T$ **do**
> >
> > > $s_t \leftarrow$ empty Hex board
> > > $h \leftarrow \{\}$
> > > **while** $s_t \notin \mathcal{Z}$ **do**
> > >
> > > > $\boldsymbol{\pi_t} \leftarrow \text{MCTS}(s_t)$ using $\theta$
> > > > $s_{t+1} \leftarrow \tau(s_t, argmax_{\boldsymbol{\pi_t}})$
> > > > store $(s_t, \boldsymbol{\pi_t})$ in $h$
> > > > $t \leftarrow t + 1$
> > >
> > > $g \leftarrow g + 1$
> > > $z \leftarrow u(s_t)$
> > > **foreach** $(s_t, \boldsymbol{\pi_t}) \in h$ **do**
> > >
> > > > **if** *player's turn* **then**
> > > > > $r_t \leftarrow z$
> > > >
> > > > **else**
> > > > > $r_t \leftarrow -z$
> > > >
> > > > add $(s_t, \boldsymbol{\pi_t}, r_t)$ to stored data
> >
> > build training dataset $M$ out of sampled data triplets
> > $\theta_{new} \leftarrow$ Update $\theta$ using SGD with $\ell$ on $M$
> > evaluate $\theta_{new}$
> > **if** $\theta_{new}$ *better than* $\theta$ **then**
> > > replace $\theta$ with $\theta_{new}$
>
> **return** Updated neural network $\theta$

**Function** `MCTS`$(s, \theta)$:

> **Parameters:** current state to evaluate $s$, neural network for simulation $\theta$
> create root node $root$ from $s$
> $t \leftarrow root$
> **while** *computational budget left* **do**
>
> > $l \leftarrow$ SELECTANDEXPAND$(t)$
> > $r \leftarrow$ SIMULATE$(l, \theta)$
> > $t \leftarrow$ BACKPROPAGATE$(t, r)$
>
> **return** Create policy $\boldsymbol{\pi}$ from tree $t$

---

