# OpenReview forum: "From Images to Connections: Can DQN with GNNs learn the Strategic Game of Hex?"
_ICLR.cc/2024/Conference — ICLR 2024 Conference Withdrawn Submission_

### Official Review · Reviewer_9axj · 2023-10-27

**Soundness:** 3 good
**Presentation:** 2 fair
**Contribution:** 2 fair
**Rating:** 3
**Confidence:** 4

**Summary:**

The paper proposes to utilize graph neural networks in conjunction with deep q-networks for learning a game-playing agent for Hex. The authors observe that convolutional neural networks (CNNs) are great at playing games such as Go but several games such as Hex are more intuitively expressed using graph representations. As a result, CNNs cannot reason about long-range dependencies two tiles could be very far apart on an image representation but be neighbors on a graph. Thus, the authors use GNNs that can better capture relational inductive biases in such games.

The authors propose two variants of their approach. The first, GraphDQN utilizes GNNs to learn embeddings that are then passed on to DQN. The second variant, GraphAra, combines GNNs  with MCTS rollouts in a manner similar to AlphaZero. The authors then conduct an extensive empirical evaluation that compares two relevant baselines: CNNs (Gao et al 2018) and U-net.

**Strengths:**

**S1.** The technical parts of the paper are well-written and easy to read. The figures are illustrative and there is enough information on Hex to understand the key parts of the contributions.

**S2.** The authors choice of GNNs is reasoned well and I agree with their hypothesis that GNNs can capture more complicated strategies for gameplay.

**S3.** The transfer performance does show promising results.

**Weaknesses:**

Unfortunately, I felt that the paper had several weaknesses most of which pertain to the empirical evaluation. Most of my questions will pertain to the same.

**W1a.** The empirical evaluation is quite confusing and not an easy-read. There is too much information to distill. For example, Fig. 6 (and the section on supervised learning) seemed a bit out of place and I am not sure what value it adds to the paper. Another example, Fig 5. plots only until board size 15x25 but Table 1 showcases performance for a single step until 25x25.

**W1b.** Table 1 compares long range dependencies across board sizes but only captures one specific pattern. While I value the significance of these results I would have thought that more different types of long-range patterns would better illustrate the strengths of GNNs being able to capture different strategies.

**W1c.** It is not clear what the authors mean by Gao et al (2018). Is it the Mohex-3HNN (as cited and mentioned in the introduction) or is it just the CNN component? If it is the latter, then I think the authors should clearly mention this. I appreciate the authors comparing the CNN/GNN in one part (Fig 5) and the (CNN + MCTS/GNN + MCTS) in another (Table 2).

**W1d.** GNNs (similar to CNNs) have a limit to their receptive field. This is related to the # of message passing steps in GNNs.

**W2.** It is not clear why the baselines were manipulated in any way. For example, "Additionally. we do not use batch-normalization ...". This makes the evaluation a bit confusing to follow. It would have been better to leave the baselines untouched as-is and compare them. I appreciate the authors trying to make their architectures as close to each other (in parameter size etc) but I think that considering the presented results where even these modfications outperform GNNs (see W3b for the gao- rationale), this does more harm than good.

**W3a.** Finally, the baselines do seem to outperform the GNN approach significantly on most of the presented problems. For example vanilla Gao and U-net outperform the GNN on the training problem and Gao outperforms the GNN on most problems. (Data on board sizes 16x16 and over is not reported).

**W3b.** I appreciate Gao- ablation showcasing the weakness of CNNs without the padding. However, I believe that it is not really a weakness of Gao. Firstly, the Hex board game itself has the padded edges to indicate the direction (left-right/top-down) to draw the edge so there is no human expert engineering done by the gao authors. Secondly, the GNN representation that the authors use seems to incorporate some human expertise and is not domain-independent (which the authors do admit to in their section on Limitations). Overall, i think this ablation does not motivate the strengths of GNNs over CNNs convincingly.

**W3c.** For Table 2, why is MoHEX-3HNN not compared? The authors state it as a SOTA method in the introduction but it is not clear if Mohex-1k is equivalent.

**Questions:**

Below are my questions to the authors. Please feel free to merge any overlapping answers.

**Q1a.** Can you please motivate why Fig. 6 is a good addition to the paper?

**Q1b.** Are there any other patterns that you are aware where GNNs would fare better?

**Q1c.** Could you please clarify my comment on W1c.

**Q1d.** Could you please comment on why the long range dependency pattern was easily identified by the GNN? Is it because the nodes are immediate neighbors in the shannon graph for this pattern? How many iterations of message passing were used in your approach?

**Q2.** Could you please comment on W2.

**Q3a.** The results from Fig 5 are missing results on board sizes 16x16 to 24x24. Do you have some data here that better highlights the transfer performance of GNNs. Since the transferable network does not need to be trained for those sizes and since some results are already reported for them (Table 1) it seems confusing as to why they were not run.  It appears that you could have showcased transfer performance much more effectively since the trend appears to favor GNNs only at the end.

**Q3c.** Can you please elaborate on my comment on W3b.

Overall, I think this is very interesting work but has been over shadowed by a limited empirical evaluation. I hope that the authors are able to address some of my questions.

---

### Official Review · Reviewer_q6Hr · 2023-10-30

**Soundness:** 2 fair
**Presentation:** 4 excellent
**Contribution:** 2 fair
**Rating:** 3
**Confidence:** 4

**Summary:**

The primary research question of the paper is whether GNNs, given their superior ability to capture complex relationships, can replace CNNs in the context of self-play reinforcement learning for strategic board games. The authors try to make a claim: GNNs could perform better than CNNs.

**Strengths:**

- The paper is well-written and the problem is well-defined
- The experiments and analyses done on Hex is thorough

**Weaknesses:**

- EXPTIME-complete problems are generally considered to be at least as hard as, if not harder than, PSPACE-complete problems because \text{P} \subseteq \text{NP} \subseteq \text{PSPACE} \subseteq \text{EXPTIME}
To my knowledge, Chess and Go has been proven to be EXPTIME-complete problems and Hex is PSPACE-complete and thus even if GNNs performs better than CNNs on Hex, we cannot tell GNNs are generally better performed in strategic board games. If the author can show the performance of GNNs comparing with models on Go (https://ai.meta.com/blog/open-sourcing-new-elf-opengo-bot-and-go-research/) or Chess. It would be more persuasive.
ref: Stefan Reisch (1981). "Hex ist PSPACE-vollständig (Hex is PSPACE-complete)". Acta Informatica (15): 167–191.
J. M. Robson (1983). "The complexity of Go". Information Processing; Proceedings of IFIP Congress. pp. 413–417.
Fraenkel, Aviezri; Lichtenstein, David (1981). "Computing a perfect strategy for n×n chess requires time exponential in n". Journal of Combinatorial Theory. Series A. 31 (2): 199–214.

- We should compare apple with apple. The authors should list the number of the parameters, the training time of Gao network and the UNet together with the GNNs when comparing their performance.

- First mover advantage is very obvious in Hex games. It would be great if the author can list the details of some examples of the tournaments.

Tiny typo: "Gao-" in the legend of Figure 5.C which I think should be "Gao" consistent with Figure 5.1.

**Questions:**

Have the authors searched the hyperparameters of UNet and Gao when comparing with GNNs?

---

### Official Review · Reviewer_HvCH · 2023-11-01

**Soundness:** 3 good
**Presentation:** 2 fair
**Contribution:** 2 fair
**Rating:** 3
**Confidence:** 3

**Summary:**

**Problem Setting**

This work investigates the difference that neural network architectures make when training policies on structured board games. It focuses specifically on the Hex setting.

**Novel Idea**

Popular methods of observation encoding for matrices often utilize the convolutional neural network. However, the inductive biases that come with such networks often do not support board game settings where non-local dependencies are prominent.

This paper proposes a DQN-style reinforcement learning framework, using a graph neural network as the encoder for the policy and the value network. The input to the graph is taken as an observation from the environment, and the embeddings are calculated via a GNN update. The embeddings are then passed into an advantage and value head to compute the Q values.

Experiments show that the GNN based model makes less errors when calculating the Q-value on a board state that requries long term dependencies. The GNN displays better transformer performance, although it performs with less accuacy on the smaller board size.

**Strengths:**

This paper presents a simple study comparing various architectures on the Hex board game. It shows that the CNN architecture has flaws when naively used on board states. The paper showcases that a GNN based representation can better evaluate the Q values of board states. The GNN approach is shown to display less overfitting and better transfer learning. The paper presents a novel examination of the usage of GNN networks in self play reinforcement learning.

**Weaknesses:**

The significance of the paper is limited as it largely presents empirical findings in a limited domain. The game of Hex is uniquely graph based, and thus the findings here are hard to transfer to other applications. A more theoretical contribution of the effect of using GNNs vs CNNs, and when to choose each one, could provide intuition that may transfer to new domains. The combination of the GNN and the self-play reinforcement learning setup are largely orthogonal and the paper does not discuss unique ideas to this intersection. The results of the paper are satisfactory but not extraordinary, as the naive baselines perform at a similar level.

**Questions:**

See above for some suggestions on improving the contribution of this paper. The idea is presented well and makes sense. Is this method able to be evaluated on other domains? The paper makes reference to the fact that GNNs have been applied to settings like Othello, but the paper does not present evaluations on these domains. Are there reliable ways to convert non-natively graph based games into a graph format? How does the size of the GNN affect performance, and are the parameter counts comparable?

---

### Author Response · Authors · 2023-11-15

Dear reviewers,

Thank you for your valuable feedback. We appreciate your feedback and see the potential of the paper. After careful consideration, we agree that they are some substantial changes needed to improve and strengthen the paper. We have decided to withdraw the paper and will rework it accordingly. Your insights have been instrumental, and we look forward to submitting an improved version in the future.

Sincerely,
the authors